# The lack of K13-propeller mutations associated with artemisinin resistance in *Plasmodium falciparum* in Democratic Republic of Congo (DRC)

**Doudou Malekita Yobi** [1] *, **Nadine Kalenda Kayiba** [2,3], **Dieudonné Makaba Mvumbi** [1], **Raphael Boreux** [4‡], **Sebastien Bontems** [4‡], **Pius Zakayi Kabututu** [1‡], **Patrick De Mol** [4], **Niko Speybroeck** [3‡], **Georges Lelo Mvumbi** [1], **Marie-Pierre Hayette** [4]

1 Department of Basic Sciences, Faculty of Medicine, University of Kinshasa, Kinshasa, Democratic Republic of Congo, 2 School of Public Health, Faculty of Medicine, University of Kinshasa, Kinshasa, Democratic Republic of Congo, 3 School of Public Health & Research Institute of Health and Society, Catholic University of Louvain, Louvain-la-Neuve, Belgium, 4 Laboratory of Clinical Microbiology, University of Liège, Liège, Belgium

☯ These authors contributed equally to this work.
‡ These authors also contributed equally to this work.
* Doudou.yobi@unikin.ac.cd

**Data Availability Statement:** All relevant data are within the manuscript and its Supporting Information files.

## Abstract

Artemisinin-based combination therapies (ACTs) have been recommended by the World Health Organization (WHO) as first-line treatment of uncomplicated *Plasmodium falciparum* (*P. falciparum*) malaria since 2005 in Democratic Republic of Congo (DRC) and a regular surveillance of the ACT efficacy is required to ensure the treatment effectiveness. Mutations in the propeller domain of the *pfk13* gene were identified as molecular markers of artemisinin resistance (ART-R). This study investigated the *pfk13*-propeller gene polymorphism in clinical isolates of *P. falciparum* collected in the DRC. In 2017, ten geographical sites across DRC were selected for a cross-sectional study that was conducted first in Kinshasa from January to March, then in the nine other sites from September to December. Dried blood samples were collected from patients attending health centers for fever where diagnosis of Malaria was first made by rapid diagnostic test (RDT) available on site (SD Bioline malaria Ag Pf or CareStart Malaria Pf) or by thick blood smear and then confirmed by a *P. falciparum* real-time PCR assay. A *pfk13*-propeller segment containing a fragment that codes for amino acids at positions 427–595 was amplified by conventional PCR before sequencing. In total, 1070 patients were enrolled in the study. Real-time PCR performed confirmed the initial diagnosis of *P. falciparum* infection in 806 samples (75.3%; 95% CI: 72.6%– 77.9%). Of the 717 successfully sequenced *P. falciparum* isolates, 710 (99.0%; 95% CI: 97.9% - 99.6) were wild-type genotypes and 7 (1.0%; 95% CI: 0.4% - 2.1%) carried non-synonymous (NS) mutations in *pfk13*-propeller including 2 mutations (A578S and V534A) previously detected and 2 other (M472I and A569T) not yet detected in the DRC. Mutations associated with ART-R in Southeast Asia were not observed in DRC. However, the presence of other mutations in *pfk13*-propeller gene calls for further investigations to assess their implication in drug resistance.

**Funding:** This work was supported by the Académie de Recherche et d'Enseignement Supérieur (ARES), Belgium.

**Competing interests:** The authors declare that they have no competing interests.

## Introduction

*Plasmodium falciparum* (*P. falciparum*) is the most widespread *Plasmodium* species and is responsible for the majority of severe forms and deaths related to malaria in sub-Saharan Africa. *P. falciparum* has succeeded in developing resistance mechanisms against almost all existing antimalarial drugs, which is a major threat to malaria control worldwide. The high level of *P. falciparum* resistance to chloroquine (CQ) and then to sulfadoxine-pyrimethamine (SP) has led endemic countries to change their antimalarial drug policy of uncomplicated malaria treatment. In the Democratic Republic of Congo (DRC), Artemisinin-based Combination Therapies (ACTs) have been adopted as first-line treatment of uncomplicated malaria in 2005 [1]. With the discovery of several *P. falciparum* genes involved in antimalarial drug resistance, molecular markers have become a precious tool in resistance surveillance [2, 3]. Mutations in the propeller domain of *P. falciparum* Kelch 13 gene (*pfk13)* have been identified as associated with in-vivo delayed parasite clearance and in-vitro artemisinin resistance (ART-R) in ring stage survival Assay (RSA) [4, 5]. These mutations spread into the Greater Mekong Sub-region (GMS) of Southeast Asia [6, 7] and have been recently classified in validated markers of ART-R (F446I, N458Y, M476I, Y493H, R539T, I543T, P553L, R561H and C580Y) and candidate/associated markers of ART-R (P441L, G449A, C469F, A481V, P527H, N537I, G538V, V568G, P574L, F673I, A675V) [8]. In Sub-Saharan Africa, there has been no evidence for emergence of *Pf* ART-R until now. Mutations detected in Africa have not been associated to ART-R like those reported in Southeast Asia [6, 7, 9–11]. Some mutations associated with slow parasite clearance detected in Africa have been reported with very low relative frequency [12–14]. The spread of ART-R from the GMS to Africa, like previously happened with other antimalarial drugs [15, 16], may be a major obstacle for malaria control and elimination around the World.

In DRC, like in almost all Sub-Saharan countries, none of validated ART-R mutations detected in Southeast Asia has been found. However, the S522C mutation which has been described to be associated with delayed clearance but without statistical significance [8], has been detected by a previous study in one isolate in DRC [17]. To continue monitoring, this study investigated the *pfk13*-propeller polymorphisms in clinical isolates of *P. falciparum* collected across the DRC.

## Materials and methods

### Study area

Ten sites were selected among the 26 provinces of the DRC. The study included the 3 largest cities of the country (Kinshasa, Kisangani and Lubumbashi) as well as 7 other sites that were selected based on their epidemiological facies of malaria: Bolenge, Karawa and Vanga for the equatorial facies, Kalima, Kamina and Fungurume for the tropical facies and Katana for the mountainous facies. All selected sites are National Malaria Control Program (NMCP) sentinel sites for malaria surveillance, including Kingasani in Kinshasa and Kabondo in Kisangani, except Lubumbashi. These sites are organized within one selected Health Zone entity, per province and are part of a national network for malaria surveillance. In each study site, 4 to 5 health structures were selected based on the accessibility and the attendance for the realization of the study. A pilot study was conducted first at Kingasani in Kinshasa from January to March 2017. Afterwards, the study was started in the other sites from September to December 2017.

### Study participants

Patients of all ages presenting at study site for fever and who had a positive rapid diagnostic test (RDT) or a positive thick blood smear for malaria were enrolled after informed consent was given.

## Blood sample collection

Screening tests were performed on blood samples taken by finger prick. The SD Bioline Malaria Ag Pf (Standard Diagnostics) assay was used to enroll patients attending all collection sites. However, in Kinshasa, the CareStart Malaria Pf (Access Bio) was also used. In case of lack of RDTs, a positive thick blood smear for malaria was used for enrollment.

After enrollment, a blood sample was taken by finger prick and three spots were deposited on Whatman Grade GB003 filter paper (Whatman/GE Healthcare). Dried blood spots (DBS) were placed in an individual ziploc plastic bag containing silicagel desiccant and were then stored at room temperature before their transfer to the laboratory of the Department of Clinical Microbiology (University of Liège) for molecular analysis.

## DNA extraction

DNA was extracted from blood spots by using the QIAamp DNA Mini Kit (Qiagen, Germany) following the recommended protocol for DBS. The extracted DNA was stored at– 20˚C before PCR testing.

## *P. falciparum* real-time PCR

A real-time PCR for the detection of *P. falciparum* was performed according to a modified procedure previously described [18], as follows: 200 nM of *P. falciparum* primers and probe, a volume of 2.5 μl of Double-Dye Probe/Primer for Internal Positive Control (IPC), 2.5 μl of DNA virus culture (DIA-EIC/DNA(Cy5) for IPC, 12.5 μl of 2X Taqman Universal PCR Master Mix (Applied Biosystems) and water to make a final volume of 25 μl including 5 μl of DNA sample. Assays were run on an ABI 7500 Fast real-time thermocycler (Applied Biosystems).

## *pfk13* PCR

A conventional PCR was developed at the Laboratory of Clinical Microbiology of University hospital of Liege using *P.falciparum* strains W2, 3D7 and IPC 3445 obtained from parasite culture and clinical isolates from patients with *P. falciparum* malaria. The PCR-amplified sequence consisted of a 506-nucleotide fragment of the *pfk13*-propeller gene (containing codons 427–595) which included recently described mutations associated with ART-R [8]. The forward primer ArtprdF: 5′-ACTGTAAAACGACGGCCAGTCCATTAGTTCCACCAATG ACA-3′ and reverse primer ArtprdR: 5′-ACCAGGAAACAGCTATGACCAGCCTTGTTGAA AGAAGCAGA-3′ were designed and used in a reaction mixture containing 0.4μM of each primer, 10μl of LightCycler$^R$ Multiplex DNA Master 5X version 6 (Roche Diagnostics GmbH, Germany), 10μl of genomic DNA, and water to make a final volume of 50μl. The PCR was run on a conventional Dyad Peltier Thermal Cycler (Bio-Rad Laboratories, CA, US) according to the following cycling parameters: Initial denaturation at 95˚ C for 9 minutes followed by 10 cycles of a denaturation at 95˚C for 30 sec, an annealing at 43˚C for 1 min, an extension at 72˚C for 1 min, followed by 35 cycles of a denaturation at 95˚C for 30 sec, an annealing at 60˚C for 1 min, an extension at 72˚C for 1 min and a final extension at 72˚C for 5 min. The PCR products were visualized after electrophoresis on 2% agarose gel stained with ethidium bromide.

## *pfk13* genotyping

After purification using AMPure XP magnetic beads (Beckman Coulter, CA, US), the PCR products were added to a mix of Big Dye Terminator V3.1 for the sequencing reaction. The resulting 506-bp nucleotide sequences were analysed on an ABI 3730 DNA Analyzer

automated sequencer (Applied Biosystems) using the Sanger method at GIGA (University of Liège's interdisciplinary research institute in the biomedical sciences). These 506-bp sequences (forward and reverse) of the *pfk13* gene encompassing the codons at position 427–595, were aligned using Vector NTI (Thermo Fisher Scientific, US) and compared to the reference sequence PF3D7_1343700 (http://www.plasmodb.org, accessed on May 6, 2019) using the online Basic Local Alignment Search Tool (BLAST) (National Center for Biotechnology Information-NCBI) for identifying mutations. Sequences have not been deposited in any data repository.

### Ethical considerations

The protocol and the informed consent received approbation from the Ethics Committee of the Faculty of Medicine, University of Kinshasa (approval N˚: ESP MINESU 019/2016). All the participants including parents or guardians of children involved in the study signed an informed consent.

### Statistical analysis

Data were entered in a 2010 Excel sheet by an independent data clerk. Statistical analysis was performed using SPSS V. 20.0 (IBM corp, Armonk, NY). Samples for which genotype profile could not be determined were excluded from the analysis. Categorical variables were expressed as frequencies with 95% confidence intervals (95% CI). The mutant and wild-type alleles identified in the sequenced isolates were used to generate the prevalence of the alleles.

## Results

A total of 1070 patients were enrolled in the study, their median age was 7 years (IQR 3–18 years) and the male to female sex ratio was 0.94.

Real-time PCR analysis performed on DNA extracted from DBS confirmed the initial diagnosis of *P. falciparum* infection for 806 (75.3%; 95% CI: 72.6%– 77.9%) patients. Of the 717 successfully sequenced *P. falciparum* isolates, 710 (99.0%; 95% CI: 97.9% - 99.6) were wild-type and 7 (1.0%; 95% CI: 0.4% - 2.1%) carried non-synonymous (NS) mutations in *pfk13*-propeller among which 2 previously described (A578S, V534A) and 2 mutations not yet described (M472I, A569T) in the DRC. A578S mutation was detected in 4 study sites while M472I, V534A and A569T were found only in one site each, as shown in Table 1.

In addition, the distribution of mutations per age range of patients showed the presence of A578S in all age ranges (Table 2).

## Discussion

The present study was conducted in ten sites across DRC for assessing the prevalence of polymorphisms in *pfk13*-propeller gene related to ART-R. We report that 99.0% of *P. falciparum* isolates were wild-type alleles and none of ART-R mutations found in Southeast Asia has been detected among the 1.0% of isolates carrying NS mutations.

Previous studies reported very limited polymorphisms in *Pfk13* ranging from 0 to 3% in the DRC and others African countries including Rwanda, Gabon, Cameroun, Republic of Congo, Angola [12, 14, 19–21]. In the present study, seven *P. falciparum* isolates carried NS mutations among which A578S in 4 isolates and A569T, M472I and V534A in 1 isolate each. In a worldwide survey published in 2016, reporting 14 NS mutations in the DRC, 3 (21.4%) A578S and 3 (21.4) V534A were characterized while A569T and M472I were not found in the DRC but reported in Rwanda (1/5) and in Niger (1/8) respectively [6]. A578S is commonly detected in Sub-Saharan African countries [6, 7, 9, 11] and has also been reported out of Africa such as in

**Table 1. Polymorphisms in *pfk13*-propeller per study site.**

| Site | Positive RDT or thick blood samples N | Positive *Pf* PCR samples N | Positive Pf PCR % (95% CI) | *Pf* PCR + successfully sequenced samples N | Mutant-type parasites N | Mutant-type parasites % (95% CI) | NS Mutation |
|---|---|---|---|---|---|---|---|
| Bolenge | 98 | 73 | 74.5 (64.7–82.8) | 63 | 1 | 1.6 (0.0–8.5) | A578S |
| Fungurume | 92 | 65 | 70.7 (60.2–79.7) | 60 | 1 | 1.7 (0.0–8.9) | A578S |
| Kalima | 97 | 67 | 69.1 (58.9–78.1) | 67 | 0 | 0.0 (0.0–5.4) | - |
| Kamina | 98 | 73 | 74.5 (64.7–82.8) | 72 | 1 | 1.4 (0.0–7.5) | A569T |
| Karawa | 97 | 66 | 68.0 (57.8–77.1) | 64 | 0 | 0.0 (0.0–5.6) | - |
| Katana | 105 | 78 | 74.3 (64.8–82.3) | 73 | 1 | 1.4 (0.0–7.4) | M472I |
| Kinshasa | 160 | 160 | 100 (100.0–100.0) | 100 | 1 | 1.0 (0.0–5.4) | A578S |
| Kisangani | 128 | 93 | 72.7 (64.1–80.2) | 92 | 1 | 1.1(0.0–5.9) | A578S |
| Lubumbashi | 101 | 56 | 55.4 (45.2–65.3) | 53 | 0 | 0.0 (0.0–6.7) | - |
| Vanga | 94 | 75 | 79.8 (70.2–87.4) | 73 | 1 | 1.4 (0.0–7.4) | V534A |
| Total | 1070 | 806 | 75.3 (72.6–77.9) | 717 | 7 | 1.0 (0.4–2.1) | |

N: Number; *Pf*: *Plasmodium falciparum*, PCR: Polymerase chain reaction, RDT: Rapid diagnostic test; NS: Non-synonymous.

Thailand and Bangladesh [22]. The survey cited above reported 41 A578S mutations in Africa out of a total of 42 worldwide. A578S is of interest for several reasons: 1) it is the most frequent mutation detected in 4 study sites (Kinshasa, Bolenge, Kisangani and Fungurume) and in all age ranges; 2) it is located 2 amino acids upstream of the C580Y Single Nucleotide Polymorphism (SNP), which is one of the mutations involved in ART-R and associated with 85% of the ART treatment failure in Southeast Asia [6]; 3) in this mutation, a neutral nonpolar amino acid (A) is replaced with a neutral but polar amino acid (S). Computational modeling and a mutational sensitivity prediction suggest that the A578S SNP could disrupt the function of *pfk13*-propeller [22]. However, some studies have shown that the A578S mutation was associated neither with slow parasite clearance nor with reduced in-vitro ART susceptibility [5, 6]. The effect of this mutation mostly detected in parasites from Africa remains unknown, additional local studies are needed to clarify the significance of this NS substitution.

Three other mutations with unknown effects were found in the present study, notably M472I, V534A and A569T. M472I has been detected in 1 isolate in Katana site located in Sud-Kivu province in the eastern part of country. In this mutation, Methionin (M) is replaced by Isoleucin (I) at position 472 in PFK13 protein. A Similar coding substitution is observed 4 amino acids downstream of an ART-R validated marker namely M476I [8]. The M472I mutation not far from M476I could also alter the function of the *pfk13* gene with implication in resistance to ART. The M476I mutation was first observed in a Tanzanian strain (F32-ART) built from parasites that were exposed in vitro to increasing concentrations of ART [5], afterward, it has been found in a patient in Tanzania [13]. The Katana site corresponds to the

**Table 2. Distribution of mutations per age range of patients.**

| NS mutation | Age range (years) | | | | Total |
|---|---|---|---|---|---|
| | 0–5 | 6–14 | 15–49 | 50–74 | |
| M472I | 1 | 0 | 0 | 0 | 1 |
| V534A | 1 | 0 | 0 | 0 | 1 |
| A569T | 0 | 0 | 1 | 0 | 1 |
| A578S | 1 | 1 | 1 | 1 | 4 |
| Total | 3 | 1 | 2 | 1 | 7 |

mountainous facies in which a lower malaria transmission rate results in drug resistant genotypes not being broken down due to less sexual recombination occurring. Drug resistance appears and emerges from this kind of places as previously shown in South-East Asia, considered as the bastion of antimalarial resistance [23]. Such a suspicious mutation found in the eastern part of DRC should be a signal for continuous surveillance because it has been suggested that in Africa, antimalarial drug resistance has historically risen in the East and spreads to the rest of the continent [24]. In addition, political instability in the East of the country is an obstacle to good management of the use of antimalarial drugs as recommended by the national policy. V534A which was previously detected in the DRC (3 out of 14 NS mutations) and A569T previously found elsewhere (1 out of 5 NS mutations in Rwanda) [6], should also call for further surveillance studies. Although the frequency of wild-type *pfk13*-propeller genes was highly predominant, we observed three coding substitutions that are of unknown phenotype. Numerous other coding substitutions of unclear phenotype have been reported in several African countries such as in Tanzania, Cameroun and Kenya [13, 25, 26]. Additional information, not assessed by the present study, such as the clinical and parasitological responses to malaria treatment in enrolled patients could help for a better interpretation of the biological and clinical impact of these mutations. There are criteria for prioritizing further laboratory studies notably the frequent observation of a new allele with a non-synonymous mutation, the evidence of dissemination and preliminary association with clinical data whenever possible [6, 7].

Several independent single nucleotide polymorphisms (SNPs) could be responsible of the ART-R in Sub-Saharan Africa, ongoing molecular surveillance will need to detect the presence of new mutations, shifts in frequencies of existing African genotypes, and the importation of Southeast-Asian resistance mutations. The known SNPs that confer drug resistance would differ from one location to another, depending on the parasite genetics. There is the possibility that *pfk13* mutations do not cause ART-R in isolation but would act in combination with other genetic or non-genetic factors that are different in African and Southeast Asia parasite populations [9, 27]. Since African *pfk13*-propeller mutations were shown to be different from those found in Southeast Asia, further molecular and biochemical studies should investigate whether other factors such as additional mutations could be associated to alter the functions of PFK13 protein, resulting in altered ART sensibility.

Routine monitoring and surveillance of ART resistance, as recommended by WHO, should be continuously strengthened to ensure the effectiveness of anti-malarial drugs in use. In the DRC, the national policy supports two first-line ACT treatments: artesunate-amodiaquine (AS/AQ) and artemether-lumefantrine (AL) for uncomplicated malaria. If one of the two first-line ACTs is not available or is poorly tolerated by the patient, the other one can be used. In case of confirmed treatment failure by microscopy to both first-line ACTs, the patient should be given dual therapy of quinine plus clindamycin. For treatment of severe malaria, injectable artesunate should be the first treatment option, followed by injectable artemether or intravenous quinine [28]. This study contributes to the ongoing surveillance of ART resistance in the DRC. However, a broader scope in *pfk13 gene* would have been desirable to explore all candidate and probable markers of ART resistance previously described [6, 7].

## Conclusion

Mutations associated with ART-R in Southeast Asia were not observed in the DRC. However, the appearance of coding substitutions that are of unknown phenotype calls for further investigations. Routine monitoring must continue in order to ensure that the recommended ACTs are effective, that timely changes to national treatment policies can be implemented, and that ART-R can be detected early.

## Supporting information

**S1 Data. Minimal data set containing supporting information (raw data) for study findings in different study parameters (site, age, sex, RDT result, P. falciparum PCR result, K13-propeller polymorphism).**
(XLSX)

**S2 Data. Nucleotide sequences for *Plasmodium falciparum* isolates successfully sequenced (wild and mutants).**
(XLSX)

## Acknowledgments

The study was conducted in collaboration with the National Malaria Control Programme (NMCP). We would like to thank the staff of all collecting sites and patients for their involvement in this study.

## Author Contributions

**Data curation:** Doudou Malekita Yobi.

**Formal analysis:** Doudou Malekita Yobi.

**Funding acquisition:** Patrick De Mol, Georges Lelo Mvumbi, Marie-Pierre Hayette.

**Investigation:** Doudou Malekita Yobi, Nadine Kalenda Kayiba, Dieudonné Makaba Mvumbi.

**Methodology:** Doudou Malekita Yobi, Nadine Kalenda Kayiba, Dieudonné Makaba Mvumbi, Raphael Boreux, Sebastien Bontems, Pius Zakayi Kabututu, Patrick De Mol, Niko Speybroeck, Georges Lelo Mvumbi, Marie-Pierre Hayette.

**Project administration:** Patrick De Mol, Georges Lelo Mvumbi, Marie-Pierre Hayette.

**Writing – original draft:** Doudou Malekita Yobi.

**Writing – review & editing:** Doudou Malekita Yobi, Nadine Kalenda Kayiba, Dieudonné Makaba Mvumbi, Raphael Boreux, Sebastien Bontems, Pius Zakayi Kabututu, Patrick De Mol, Niko Speybroeck, Georges Lelo Mvumbi, Marie-Pierre Hayette.

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
