## [Decision Letter · Decision Letter 0]

6 May 2020

PONE-D-20-05168

K13-propeller gene polymorphisms in Plasmodium falciparum: the lack of Artemisinin resistance markers in the Democratic Republic of Congo

PLOS ONE

Dear Dr Yobi,

Thank you for submitting your manuscript to PLOS ONE. After careful consideration, we feel that it has merit but does not fully meet PLOS ONE’s publication criteria as it currently stands. Therefore, we invite you to submit a revised version of the manuscript that addresses the points raised during the review process.

We would appreciate receiving your revised manuscript by Jun 20 2020 11:59PM. To enhance the reproducibility of your results, we recommend that if applicable you deposit your laboratory protocols in protocols.io, where a protocol can be assigned its own identifier (DOI) such that it can be cited independently in the future. For instructions see: http://journals.plos.org/plosone/s/submission-guidelines#loc-laboratory-protocols

We look forward to receiving your revised manuscript.

Kind regards,

Md Atique Ahmed, PhD

Academic Editor

PLOS ONE

Additional Editor Comments (if provided):

Please prepare a revised version of the manuscript addressing the concerns raised by the experts.

2. We note that Figure 1 in your submission contains map images which may be copyrighted. All PLOS content is published under the Creative Commons Attribution License (CC BY 4.0), which means that the manuscript, images, and Supporting Information files will be freely available online, and any third party is permitted to access, download, copy, distribute, and use these materials in any way, even commercially, with proper attribution. For these reasons, we cannot publish previously copyrighted maps or satellite images created using proprietary data, such as Google software (Google Maps, Street View, and Earth). For more information, see our copyright guidelines: http://journals.plos.org/plosone/s/licenses-and-copyright.

Reviewers' comments:

Reviewer's Responses to Questions

**Comments to the Author**

1. Is the manuscript technically sound, and do the data support the conclusions?

Reviewer #1: Yes

Reviewer #2: Yes

2. Has the statistical analysis been performed appropriately and rigorously? 

Reviewer #1: N/A

Reviewer #2: Yes

3. Have the authors made all data underlying the findings in their manuscript fully available?

Reviewer #1: Yes

Reviewer #2: Yes

4. Is the manuscript presented in an intelligible fashion and written in standard English?

Reviewer #1: Yes

Reviewer #2: Yes

5. Review Comments to the Author

Reviewer #1: - The author studied the K13 gene mutation in falciparum malaria in Congo. It is an important topic for respective field. It is needed to revised to improve the quality of the paper.

- Tile: Consider to revise the title and running title to cover the whole picture of the study. Eg, Lack of K13….. in Congo. Running title: Pf K13 polymorphism in DROC.

- Line 43, M472I, V534A and A569T are not new mutation. These are already reported. (DOI: 10.9734/JAMMR/2017/34192) https://www.ncbi.nlm.nih.gov/pmc/articles/PMC4955562/ (Major comment) If you want to say these are new SNPs detected in DRC, clearly mention in abstract also.

- The author use the term "novel mutation" for SNP detected that werenot reported before in DRC. But these SNPs are already reported in other areas. It mean it is not novel mutation. Correct the term used in the paper. (Figure also for "new mutation")

- In discussion, among the previous reported SNP, number as well as percentage should be shown.

- The authors also used the term "New developed pfk13 PCR" for newly designed primer. It is also not need to mention the term "new developed" that is for new innovative procedures that is not reported elsewhere.

- Here again, why the authors do not use the previously published primers set? Describe it. Moreover, the primers used in this study cover only codons 427 – 595. So, it will miss the SNP beyond the aa 595 in K13 propeller gene. A676 and H719 were also interesting SNPs and reported. The authors need to describe the rationale for it.

- It is better to include the basic demographic character of the participants as well as the clinical data of day3 positivity (if possible)

- Describe the first line and second line treatment for pf in Congo in Discussion.

Reviewer #2: 1. Summary of the research

Yobi et.al have looked at polymorphisms in the Kelch 13 propeller gene in P.falciparum isolates from clinical samples in the DRC and have reported both existing and new non-synonymous mutations in the gene from the region; although none of these mutations have been directly implicated in ART resistance. The study has been conducted with comprehensive coverage of the major malaria endemic regions of the country with an adequate sample size. The diagnostics and methodology are robust and the results are expressed in a logical manner. Given the fact that artemisinin is the mainstay for treatment for uncomplicated falciparum malaria, monitoring its activity is important in the field and in this regard, the authors have looked into K-13 mutations which is a preliminary molecular marker for possible artemisinin resistance.

2. Examples and evidence

Major issues

1. The authors have looked into K-13 polymorphisms within a limited codon range (427-595). Although most of the validated K-13 propeller mutations associated with ART resistance are covered in this range, a broader scope could have been adopted to cover other candidate mutations. Since ART resistance is still evolving in Africa, a comprehensive investigation into all established and probable markers of ART resistance would have given a clearer picture of the situation.

2. The authors have used a Taqman based qPCR assay for P.falciparum detection. As such, it would be good if qPCR based parasite quantification data is mentioned in the results. It would be especially interesting to learn about the difference and correlation (if any) in the parasite counts in individuals harbouring wild-type versus mutant K-13 isolates.

3. No mention is made of the patient demographics in the manuscript. Any possible association between patient demographics and isolation of mutant parasite strains should be looked into and included in the results.

Minor issues

1. Since drug resistance mutations often have a tendency to cluster together, it would be good to know if the isolates harbouring K-13 mutations also have other antimalarial drug mutations. If these isolates have been subjected to such tests, the information should be included in the results.

2. The sequenced isolates do not mention any accession numbers. If these sequences have not been deposited in any data repository, the information should be included in the methodology.

6. PLOS authors have the option to publish the peer review history of their article (what does this mean?). If published, this will include your full peer review and any attached files.

Reviewer #1: Yes: Myat Htut Nyunt

Reviewer #2: No

---

## [Author Response · Author response to Decision Letter 0]

11 Jun 2020

Responses to Reviewers

Reviewer #1: 

We thank the reviewer very much for the comments, which helped us improving the manuscript.

- Tile: Consider to revise the title and running title to cover the whole picture of the study. Eg, Lack of K13….. in Congo. Running title: Pf K13 polymorphism in DROC.

A/ Thank you for the suggestion. We’ve revised the title and running title.

- Line 43, M472I, V534A and A569T are not new mutation. These are already reported. (DOI: 10.9734/JAMMR/2017/34192) https://www.ncbi.nlm.nih.gov/pmc/articles/PMC4955562/ (Major comment) If you want to say these are new SNPs detected in DRC, clearly mention in abstract also.

A/ Thank you for pointing this out. We’ve corrected in the manuscript: we’ve removed the term "new" and preferred say "previously detected and not yet detected in DRC"

- The author use the term "novel mutation" for SNP detected that were not reported before in DRC. But these SNPs are already reported in other areas. It means it is not novel mutation. Correct the term used in the paper. (Figure also for "new mutation")

A/ Thank you for the remark. We’ve corrected in the whole text.

- In discussion, among the previous reported SNP, number as well as percentage should be shown.

A/ Thank you for the suggestion. We’ve added number and percentage for previous reported SNP.

- The authors also used the term "New developed pfk13 PCR" for newly designed primer. It is also not need to mention the term "new developed" that is for new innovative procedures that is not reported elsewhere.

A/ Thank you for the remark. We’ve removed the term "New"

- Here again, why the authors do not use the previously published primers set? Describe it. Moreover, the primers used in this study cover only codons 427 – 595. So, it will miss the SNP beyond the aa 595 in K13 propeller gene. A676 and H719 were also interesting SNPs and reported. The authors need to describe the rationale for it.

A/ Thank you for the comment. We’ve designed new primers to amplify a segment containing all validated mutations and majority of candidate mutations associated to artemisinin resistance. We’ve taken into account the limit size (700-800bp) of segment to ensure a better quality of sequences obtained from Sanger sequencing method used in this study. A broader view would have been desirable to explore other candidate mutations. We’ve mentioned this limitation in discussion section.

- It is better to include the basic demographic character of the participants as well as the clinical data of day3 positivity (if possible)

A/ Thank you for the suggestion. We’ve included in the results the repartition of participants per age range and per sex. Clinical follow-up data were not assessed by the study. 

- Describe the first line and second line treatment for pf in Congo in Discussion.

A/ Thank you for the suggestion. We’ve include an overview of first-line and second-line treatment used in DRC for uncomplicated malaria in the discussion section. 

Reviewer #2: 

1. Summary of the research

Yobi et.al have looked at polymorphisms in the Kelch 13 propeller gene in P.falciparum isolates from clinical samples in the DRC and have reported both existing and new non-synonymous mutations in the gene from the region; although none of these mutations have been directly implicated in ART resistance. The study has been conducted with comprehensive coverage of the major malaria endemic regions of the country with an adequate sample size. The diagnostics and methodology are robust and the results are expressed in a logical manner. Given the fact that artemisinin is the mainstay for treatment for uncomplicated falciparum malaria, monitoring its activity is important in the field and in this regard, the authors have looked into K-13 mutations which is a preliminary molecular marker for possible artemisinin resistance.

We sincerely thank the reviewer for these very encouraging comments and remarks to improve this work.

2. Examples and evidence

Major issues

1. The authors have looked into K-13 polymorphisms within a limited codon range (427-595). Although most of the validated K-13 propeller mutations associated with ART resistance are covered in this range, a broader scope could have been adopted to cover other candidate mutations. Since ART resistance is still evolving in Africa, a comprehensive investigation into all established and probable markers of ART resistance would have given a clearer picture of the situation.

A/ Thank you for the comment. As you said, we’ve amplified a segment containing all validated mutations and majority of candidate mutations associated to artemisinin resistance. We’ve taken into account the limit size (700-800bp) of segment to ensure a better quality of sequences obtained from Sanger sequencing method used in this study. A broader view would have been desirable to explore other candidate mutations. We’ve mentioned this limitation in discussion section.

2. The authors have used a Taqman based qPCR assay for P.falciparum detection. As such, it would be good if qPCR based parasite quantification data is mentioned in the results. It would be especially interesting to learn about the difference and correlation (if any) in the parasite counts in individuals harbouring wild-type versus mutant K-13 isolates.

A/ Thank you for the suggestion. In this study, we did not quantify parasite densities in participants, we’ve just noted the presence or not of the Plasmodium falciparum. We will take this suggestion into account in our future studies.

3. No mention is made of the patient demographics in the manuscript. Any possible association between patient demographics and isolation of mutant parasite strains should be looked into and included in the results.

A/Thank you for the suggestion. We’ve included in results the distribution of patients per age range and per sex and the distribution of mutations per age range of patients.

Minor issues

1. Since drug resistance mutations often have a tendency to cluster together, it would be good to know if the isolates harbouring K-13 mutations also have other antimalarial drug mutations. If these isolates have been subjected to such tests, the information should be included in the results.

A/ Thank you for pointing this out. Other antimalarial drug mutations including mutations on pfcrt gene associated with chloroquine resistance were addressed in another paper.

2. The sequenced isolates do not mention any accession numbers. If these sequences have not been deposited in any data repository, the information should be included in the methodology.

A/ Thank you for pointing this out. We’ve mentioned in Methods section the sequences were not deposited in any data repository.

---

## [Decision Letter · Decision Letter 1]

14 Jul 2020

PONE-D-20-05168R1

The lack of K13-propeller mutations associated with artemisinin resistance in Plasmodium falciparum in Democratic Republic of Congo (DRC)

PLOS ONE

Dear Dr. Yobi,

Thank you for submitting your manuscript to PLOS ONE. After careful consideration, we feel that it has merit but does not fully meet PLOS ONE’s publication criteria as it currently stands. Therefore, we invite you to submit a revised version of the manuscript that addresses the points raised during the review process.

I would request the authors to revise the Table 1 as suggested by reviewer 1 before a final decision can be is taken

We look forward to receiving your revised manuscript.

Kind regards,

Md Atique Ahmed, PhD

Academic Editor

PLOS ONE

Additional Editor Comments (if provided):

Please address Reviewer 1 comments before a final decision is taken.

Reviewers' comments:

Reviewer's Responses to Questions

**Comments to the Author**

1. If the authors have adequately addressed your comments raised in a previous round of review and you feel that this manuscript is now acceptable for publication, you may indicate that here to bypass the “Comments to the Author” section, enter your conflict of interest statement in the “Confidential to Editor” section, and submit your "Accept" recommendation.

Reviewer #1: All comments have been addressed

Reviewer #2: All comments have been addressed

2. Is the manuscript technically sound, and do the data support the conclusions?

Reviewer #1: Yes

Reviewer #2: Yes

3. Has the statistical analysis been performed appropriately and rigorously? 

Reviewer #1: Yes

Reviewer #2: Yes

4. Have the authors made all data underlying the findings in their manuscript fully available?

Reviewer #1: Yes

Reviewer #2: Yes

5. Is the manuscript presented in an intelligible fashion and written in standard English?

Reviewer #1: Yes

Reviewer #2: Yes

6. Review Comments to the Author

Reviewer #1: Table 1. Efficacy should be "frequency or number of the cases" If you have no hypothesis on ART based on sex, you should not compare between male and female. If you have hypothesis, you can compare and statistical analysis will be needed. For my opinion, you can simply describe basic demographic character of the participants (eg Age mean(SD), M:F, region etc only.

Reviewer #2: (No Response)

7. PLOS authors have the option to publish the peer review history of their article (what does this mean?). If published, this will include your full peer review and any attached files.

Reviewer #1: No

Reviewer #2: No

---

## [Author Response · Author response to Decision Letter 1]

17 Jul 2020

Reviewer #1: 

We sincerely thank the reviewer for the suggestion, raised to improve the quality of the manuscript.

Q/ Table 1. Efficacy should be "frequency or number of the cases" If you have no hypothesis on ART based on sex, you should not compare between male and female. If you have hypothesis, you can compare and statistical analysis will be needed. For my opinion, you can simply describe basic demographic character of the participants (eg Age mean(SD), M:F, region etc only

A/ Based on the suggestion by the reviewer, we’ve opted to describe basic demographic characters of the participants (median age, sex ratio) only in the text of the manuscript in place of the table. The distribution of K13 polymorphim per region (site) of participants can be found in table 1 of the newly revised version.

---

## [Editor Report · Decision Letter 2]

4 Aug 2020

The lack of K13-propeller mutations associated with artemisinin resistance in Plasmodium falciparum in Democratic Republic of Congo (DRC)

PONE-D-20-05168R2

Dear Dr. Yobi,

We’re pleased to inform you that your manuscript has been judged scientifically suitable for publication and will be formally accepted for publication once it meets all outstanding technical requirements.

Kind regards,

Md Atique Ahmed, PhD

Academic Editor

PLOS ONE
---

## [Editor Report · Acceptance letter]

11 Aug 2020

PONE-D-20-05168R2 

The lack of K13-propeller mutations associated with artemisinin resistance in *Plasmodium falciparum* in Democratic Republic of Congo (DRC) 

Dear Dr. Yobi:

I'm pleased to inform you that your manuscript has been deemed suitable for publication in PLOS ONE. Congratulations! Your manuscript is now with our production department. 

Kind regards, 

on behalf of

Dr. Md Atique Ahmed 

Academic Editor

PLOS ONE